# Romantic partner embraces reduce cortisol release after acute stress induction in women but not in men

Gesa Berretz[1], Chantal Cebula[1], Blanca Maria Wortelmann[1], Panagiota Papadopoulou[1], Oliver T. Wolf[2], Sebastian Ocklenburg[1,3,4], Julian Packheiser[1,5]*

1 Institute of Cognitive Neuroscience, Biopsychology, Faculty of Psychology, Ruhr University Bochum, Bochum, Germany, 2 Institute of Cognitive Neuroscience, Cognitive Psychology, Faculty of Psychology, Ruhr University Bochum, Bochum, Germany, 3 Department of Psychology, Medical School Hamburg, Hamburg, Germany, 4 ICAN Institute for Cognitive and Affective Neuroscience, Medical School Hamburg, Hamburg, Germany, 5 Social Brain Lab, Netherlands Institute for Neuroscience, Amsterdam, the Netherlands

* j.packheiser@nin.knaw.nl

## Abstract

Stress is omnipresent in our everyday lives. It is therefore critical to identify potential stress-buffering behaviors that can help to prevent the negative effects of acute stress in daily life. Massages, a form of social touch, are an effective buffer against both the endocrinological and sympathetic stress response in women. However, for other forms of social touch, potential stress-buffering effects have not been investigated in detail. Furthermore, the possible stress-buffering effects of social touch on men have not been researched so far. The present study focused on embracing, one of the most common forms of social touch across many cultures. We used a short-term embrace between romantic partners as a social touch intervention prior to the induction of acute stress via the Socially Evaluated Cold Pressor Test. Women who embraced their partner prior to being stressed showed a reduced cortisol response compared to a control group in which no embrace occurred. No stress-buffering effect could be observed in men. No differences between the embrace and control group were observed regarding sympathetic nervous system activation measured via blood pressure or subjective affect ratings. These findings suggest that in women, short-term embraces prior to stressful social situations such as examinations or stressful interviews can reduce the cortisol response in that situation.

## Introduction

Embracing is one of the most prevalent forms of social touch in everyday life and across cultures [1]. They are used to greet others, convey love and affection but can also be used in negative situations to console people who feel sad or depressed [2]. Embraces have been demonstrated to be influenced by the affective state of the embracing individuals [3, 4]. Moreover, embracing your partner elicits stronger positive emotional responses both on the

**Data Availability Statement:** Ethical approval was not provided by the participants to make the data publicly available. The authors therefore are not allowed to upload the data to a public repository as

they contain demographic data that allows for potential identification of the participants. For questions regarding the ethics approval, please contact the relevant ethics committee (ethikkommission-psychologie@rub.de). To make the data accessible, we uploaded both data and code to a private repository in Open Science Framework (https://osf.io/8gyr2/). Requests for the data can be made to two of the authors (gesa. berretz@rub.de or j.packheiser@nin.knaw.nl) or to the local ethics committee of the Psychology Department at Ruhr University Bochum (ethikkommission-psychologie@rub.de).

**Funding:** O.T.W. is financially supported by the Deutsche Forschungsgemeinschaft (project number: 400672603; grant: WO733/18-1). S.O. is financially supported by the Deutsche Forschungsgemeinschaft (project number: 400672603, OC127/9-1). J.P. was financially supported by the German National Academy of Sciences Leopoldina (LPDS 2021-05) and received his salary from this organization. The funders had no role in study design, data collection and analysis, decision to publish, or preparation of the manuscript. We acknowledge support by the DFG Open Access Publication Funds of the Ruhr-Universität Bochum.

**Competing interests:** The authors have declared that no competing interests exist.

behavioral and neurophysiological level compared to embracing objects [5]. The positive feeling of an embrace is also associated with the duration of the embrace since longer embraces are perceived as more pleasant compared to very short, i.e. 1s long, embraces [6].

The positive emotional effects of embracing might explain why it is linked to benefits on physical and mental health. For example, embracing has been shown to reduce blood pressure [7], is associated with decreases in inflammation [8] and with increased subjective well-being [9]. Furthermore, embracing is linked to a reduced risk of infection and an accelerated recovery from viral diseases [10]. The latter study suggested that this health benefit might be due to the potential buffering effects of embraces on the stress response as this effect was especially dominant on days with interpersonal tension.

Stress is one of the leading factors associated with a variety of mental disorders such as depression, burn-out or anxiety disorders [11–13]. While these disorders are generally linked to long-term stress exposure, acute stress has been, amongst other things, associated with increases in negative mood indicating the importance of regulating stress for subjective well-being as well [14]. On the physiological level, the stress response comprises two separate but interconnected branches. The activation of the sympathetic nervous system leads to a fast release of adrenalin from the adrenal medulla as well as noradrenaline from postganglionic neurons resulting in an increase of heart rate, blood pressure and breathing frequency [15]. This release is paralleled by the nucleus coeruleus leading to an increase of adrenalin and noradrenalin in the brain which is connected to the subjective feeling of stress [16]. The other branch consists of the hypothalamic-pituitary-adrenocortical (HPA) axis. Here, secretion of corticotropin-releasing-hormone and arginine-vasopressin from the hypothalamic paraventricular nucleus causes the release of adrenocorticotropic hormone from the pituitary into the circulatory system. This leads to a release of glucocorticoid hormones from the adrenal cortex. In humans, cortisol represents the main glucocorticoid and is therefore regarded as one of the major stress hormones [17].

Major stressors that have been demonstrated to activate these two systems are pressure to perform, uncontrollability and social evaluation [18, 19]. While confrontation with other people can elicit a stress reaction, the presence and the support of others, especially through social touch, can also act as a buffer towards the bodily stress response. For example, Heinrichs et al. [20] exposed men to the Trier Social Stress Test (TSST), a stress induction paradigm exerting psychosocial stress [21], and found that a preparation for the social evaluation of TSST with social support reduced the secretion of cortisol. Furthermore, they also administered oxytocin which has been demonstrated to exert stress-attenuating effects [22] and is released especially through social touch [23]. Heinrichs et al. [20] found that social support plus oxytocin provided an even stronger buffering effect for the secretion of cortisol compared to social support by itself.

Prolonged social touch and the associated increased levels of oxytocin have been suggested to induce a shift from sympathetic to parasympathetic activation in animal studies [24] and have been demonstrated to increase psychophysiological relaxation in women [25]. Furthermore, embracing a human-shaped cushion during a phone conversation has been shown to reduce cortisol levels in women compared to a phone call without the embrace [26]. However, these studies did not perform a stress induction calling into question whether these effects apply to acute physiological stress reactions. To our knowledge, only a few selected studies investigated the role of prolonged physical contact on the acute stress response in the laboratory. For example, Ditzen et al. [27] studied the effects of different types of couple interactions on cortisol and heart rate responses elicited via the TSST in women. In three experimental groups, they tested the effects of no interactions between romantic partners, verbal social support and physical social support in form of a 10-minute-long shoulder massage on the

sympathetic nervous system and the HPA axis. They found significant decreases of both heart rate and cortisol release exclusively in the physical support group suggesting that physical contact provides the most effective buffer against the bodily stress response. In line with these findings, Grewen et al. [28] compared cohabitating couples that either held hands for a prolonged period (10min) and shortly embraced each other afterwards (20s) or did not touch each other prior to a public-speaking task. They found that the mutual physical contact attenuated systolic, diastolic blood pressure and heart rate increases compared to the control group. Finally, Pauley et al. [29] subjected romantic couples or platonic friends to acute stress after they talked either about fond mutual memories for a period of 10 minutes and embraced afterwards (< 10s duration), only shared each other's presence or waited separately for the stress induction to begin. The authors observed an attenuated cardiovascular stress response in all participants that went through the affectionate communication plus embracing condition. For HPA-axis activation, the findings were moderated by the relationship type as platonic friends showed a stronger increase in cortisol after expressing mutual affection compared to romantic partners that only shared each other's presence.

Since the previous studies either used long-lasting touch such as massages or combined embraces with other means such as hand holding or affectionate communication, the present study investigated if short embraces by themselves can buffer against the stress reaction since embraces have been indicated to be a viable stress buffer in stressful situations [10]. To this end, we invited romantic couples and randomly attributed them to two experimental groups. One group was asked to perform a mutual embrace before undergoing a joint stress induction procedure. The other group did not embrace each other, and partners only provided support through their physical presence. We hypothesized that the embracing group demonstrates a reduced cortisol and blood pressure response to the stressor in accordance with previous findings [25–29].

## Methods

### Participants

A total of 76 participants with an average age of 22.30 years (standard deviation: 2.24; range: 19 to 32 years) were included in the final sample. 36 participants were male and 40 were female. Participants were invited as romantic couples and therefore always came in pairs. All participants were in heterosexual relationships even though we did not exclude couples of other sexual orientations from participating in the study. The numerical imbalance in the groups resulted from post hoc exclusion of participants due to technical issues. The participants included in the final sample had no history of mental or neurodevelopmental disorders, were non-smokers, did not take any medication and had no prior experience with the experimental paradigm. Furthermore, the body mass index was between 18.5 and 27 since obesity has been linked to systematic increases in HPA-axis responsivity [30]. The study was approved by the local ethics committee at the faculty of psychology at Ruhr University Bochum. All participants gave written informed consent and were treated in accordance with the declaration of Helsinki.

### Experimental procedure

All experiments were conducted between 1pm to 6pm to control for effects of the circadian rhythm on cortisol [31]. To prevent the contamination of saliva samples, the participants were explicitly asked not to eat or drink anything apart from water for one hour before the procedure. Couples arrived jointly and were then attributed randomly to one of two experimental groups. In the control condition, couples underwent a stress induction procedure (see below

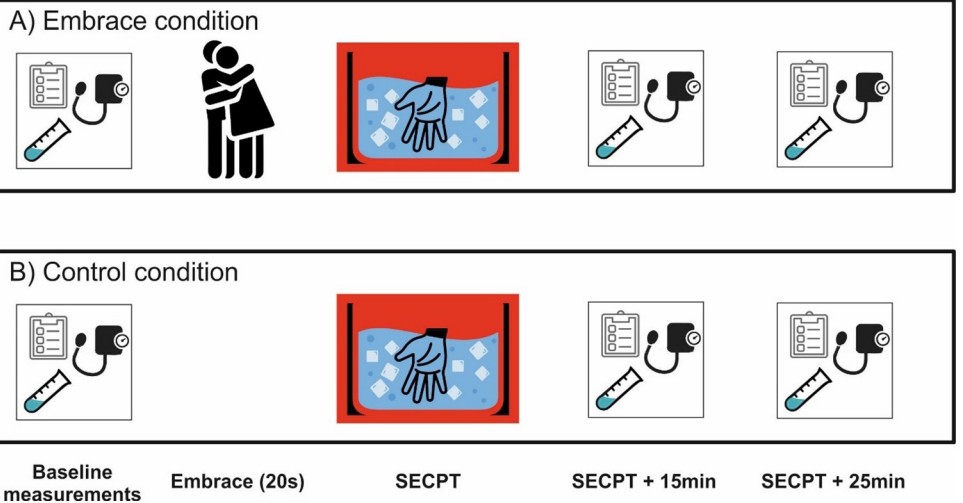

**Fig 1. A)** In the embrace condition, participants filled out questionnaires and provided a baseline measurement for cortisol, blood pressure and affective state prior to the experimental onset. After this was concluded, a 20s embrace in the absence of the experimenter took place. Following the embrace, the couples were subjected to the SECPT in a group setting with another measurement cycle of cortisol, blood pressure and affective state. 15 and 25 minutes after the SECPT was concluded, another measurement was conducted. **B)** Same as in **A)**, but the partners did not embrace prior to the SECPT.

for detail) together. In the embrace condition, participants went through the identical procedure but also embraced each other once during the experiment. In total, 19 women and 18 men were part of the control group and 21 women and 18 men were part of the embracing group.

After being picked up by the experimenter and being seated in the experimental room, participants in both groups first provided informed written consent. After filling out demographic questionnaires, a baseline saliva sample as well as systolic and diastolic blood pressure were taken (see Fig 1). Furthermore, participants filled out the Positive and Negative Affect Scale (PANAS, [32]). Following baseline measurement, the experimenter either left the room immediately to allow for the stress induction to commence (control condition) or instructed the participants to embrace in a standing position once they left the room and until they re-entered the room (embrace condition). No specific instruction was given to allow for a natural experience of the embrace. The experimenter then left and waited for exactly 20s after which they re-entered the room to signal the end of the embrace. As in the control group, the experimenter then left and the stress induction procedure began.

In both groups, a previously unknown experimenter (randomly chosen to be female or male) entered the room dressed in a lab coat. This experimenter was briefed to act neutral and distanced to the participants both during the instructions and during the stress procedure itself. The participants were then informed about the stress procedure, namely the Socially Evaluated Cold Pressor Test (SECPT) which has been evaluated for group testing scenarios [33]. The SECPT requires participants to place their hand with spread fingers in an ice-cold water bath (0–4°C) for a maximum of three minutes upon which the experimenter ends the procedure. Participants were instructed that they could remove their hand if the procedure could no longer be endured at any time. During the SECPT, participants were furthermore filmed by a camera to which they were asked to constantly hold eye contact. They were also asked not to talk during the procedure. Any violations of these instructions were continuously remarked by the experimenter. After one minute of the SECPT start, the blood pressure was

taken again, regardless of whether the procedure was aborted or not. After the procedure had ended, another saliva sample was immediately taken, and the participants filled out another PANAS questionnaire as well as subjective stress questionnaires about the SECPT. Here, the participants had to indicate on a scale from 0–100 how difficult, painful, stressful and unpleasant the SECPT procedure had been. The experimenter for the SECPT then left the room and the original experimenter came back.

Following the SECPT, the couples were separated from each other to fill out the Relationship Assessment Scale (RAS, [34]). This was done to prevent any influence on the rating by the romantic partners by having to fill out the questionnaire next to them. The RAS is a 7-item questionnaire that assesses the relationship quality on a 5-point Likert scale. Furthermore, both 15 minutes and 25 minutes after the conclusion of the SECPT, another saliva sample, blood pressure and affective ratings using the PANAS questionnaire were taken from the participants. Thus, participants provided four measurements in total (baseline, SECPT, 15min and 25min post SECPT).

## Cortisol measurements

Saliva was collected polypropylene micro salivettes (Sarstedt, Nürmbrecht, Germany). After the end of the experimental session, the samples were stored at -20˚C until further analysis. Cortisol was assessed in an in-house laboratory of the Genetic Psychology and Cognitive Psychology Department at Ruhr University Bochum by using a cortisol enzyme-linked immunosorbent assay (Cortisol Saliva ELISA, IBL, Hamburg, Germany). The intra-assay coefficients of variance (CV) were below 5% and inter-assay CVs were below 15%.

## Cardiovascular measurements

Blood pressure was taken on the left arm using an M700 Intelli IT blood pressure monitor (OMRON, Kyoto, Japan). The left arm was used as the participants were instructed to submerge their right hand under water. The cuff was situated approximately 2 cm above the elbow bend. Participants were asked to rest their arm on the table and remain silent during the measurement to ensure valid results.

## Statistical analyses

Statistical evaluation was performed using R version 4.1.0. Relationship satisfaction was compared between both groups using independent samples t-tests. Cortisol data was log-transformed prior to further analysis. We used a linear mixed model analysis using the lme4 and lmerTest package [35, 36] as these models can include subjects as a random factor in addition to fixed effects allowing for better generalization to the underlying population [37]. The model was run with (1) cortisol, (2) systolic blood pressure, (3) diastolic blood pressure as well as (4) positive and (5) negative affective state as dependent variables and measurement time point (levels: baseline, SECPT, 15min post SECPT, 25min post SECPT), condition (levels: control, embracing) and sex (levels: women, men) as independent variables. For cortisol, we also included the use of oral contraceptives (OCs) as a covariate due to its known effect of cortisol. Post-hoc tests were performed using the emmeans function to correct for multiple comparisons (Bonferroni method). A post-hoc sensitivity analysis was conducted to determine how much power would have been necessary given the present sample size to detect a within-between interaction effect using g*Power [38]. Here, we used four groups, four repeated measures, a repeated measures correlation of $r = 0.7$ and non-sphericity correction of $\varepsilon = 0.62$ as inputs. 80% power would have been achieved at interaction effects greater than Cohen's $f = 0.14$. Data that was only obtained during the SECPT, i.e., difficulty, painfulness,

stressfulness, and unpleasantness of the SECPT as well as the duration of the hand in the water were evaluated using a linear model with each of these parameters as dependent variable and group and sex as independent variables.

## Results

### Relationship satisfaction

Relationship satisfaction in the overall sample was high with a mean RAS score of 30.88 (SD = ± 3.99, maximum score = 35). The two experimental groups showed no difference in relationship satisfaction ($t_{(73)}$ = 0.54, $p$ = .593). There was furthermore no difference in relationship satisfaction between women and men ($t_{(73)}$ = 0.16, $p$ = .872).

### Cortisol

We observed significant main effects of measurement time point at 15 min and 25 min post SECPT ($\beta$ = 0.53 [0.38–0.67], SE = 0.07, $t$ = 7.12, $p < .001$; $\beta$ = 0.48 [0.33–0.62], SE = 0.07, $t$ = 6.42 $p < .001$, respectively). Bonferroni corrected post hoc tests revealed that cortisol levels were increased at 15 min and 25 min post SECPT compared to baseline and the SECPT (all $p$'s < .001) indicating that the SECPT overall succeeded in inducing a cortisol response. We also found a significant main effect of sex with men showing a higher cortisol response overall ($\beta$ = 0.37 [0.09–0.65], SE = 0.14, $t$ = 2.59, $p$ = .012). Finally, we observed a three-way interaction between condition, sex and the measurement time point at 15 and 25min post SECPT ($\beta$ = 0.61 [0.07–1.15], SE = 0.28, $t$ = 2.16, $p$ = .032; $\beta$ = 0.81 [0.27–1.35], SE = 0.28, $t$ = 2.82, $p$ = .005, respectively). For differences between the embracing and control groups, we found a significant difference only at 25min post SECPT after Bonferroni correction. Here, post-hoc tests revealed significantly less cortisol concentrations in individuals in the embracing group compared to the control group for women only ($t$ = 2.21, $p$ = .028, see Fig 2). For differences between sexes, we only found significant differences between women and men in the embracing group. Here, women demonstrated significantly lower cortisol levels compared to men at both 15min ($t$ = 2.32, $p$ = .023) and 25min post SECPT ($t$ = 2.34, $p$ = .022). Usage of OCs as a covariate did not reach significance in the model ($\beta$ = -0.15 [-0.58–0.09], SE = 0.21, $t$ = 0.71, $p$ = .480). Descriptive values for cortisol measurements across groups can be found in Table 1. The effect size of the model with and without interactions can be found in S1 Table.

### Blood pressure

For systolic blood pressure, we found a significant main effect of measurement time point during the SECPT ($\beta$ = 12.83 [9.53–16.11], SE = 1.68, $t$ = 7.62, $p < .001$) and at both the 15min ($\beta$ = -5.47 [-8.76 – -2.18], SE = 1.68, $t$ = 3.25 $p < .001$) and 25min post SECPT measurement ($\beta$ = -7.03 [-10.32– -3.74], SE = 1.68, $t$ = 4.18, $p < .001$). Bonferroni corrected post-hoc test revealed a significant increase in systolic blood pressure during the SECPT compared to baseline ($t$ = 7.62, $p < .001$), but significant decreases 15min ($t$ = 3.25 $p < .001$) and 25min after the SECPT ($t$ = 4.18, $p < .001$) compared to baseline (see S1 Fig). The model for sex also reached significance ($\beta$ = 14.79 [10.07–19.52], SE = 2.41, $t$ = 6.13, $p < .001$) with men showing higher systolic blood pressure compared to women. No other main effect or interaction reached significance (all $p$'s. > .228). Descriptive values for systolic blood pressure measurements across groups can be found in Table 2. The effect size of the model with and without interactions can be found in S1 Table.

For diastolic blood pressure, we only found a significant main effect of measurement time point during the SECPT ($\beta$ = 12.55 [9.43–15.68], SE = 1.60, $t$ = 7.86, $p < .001$). Bonferroni

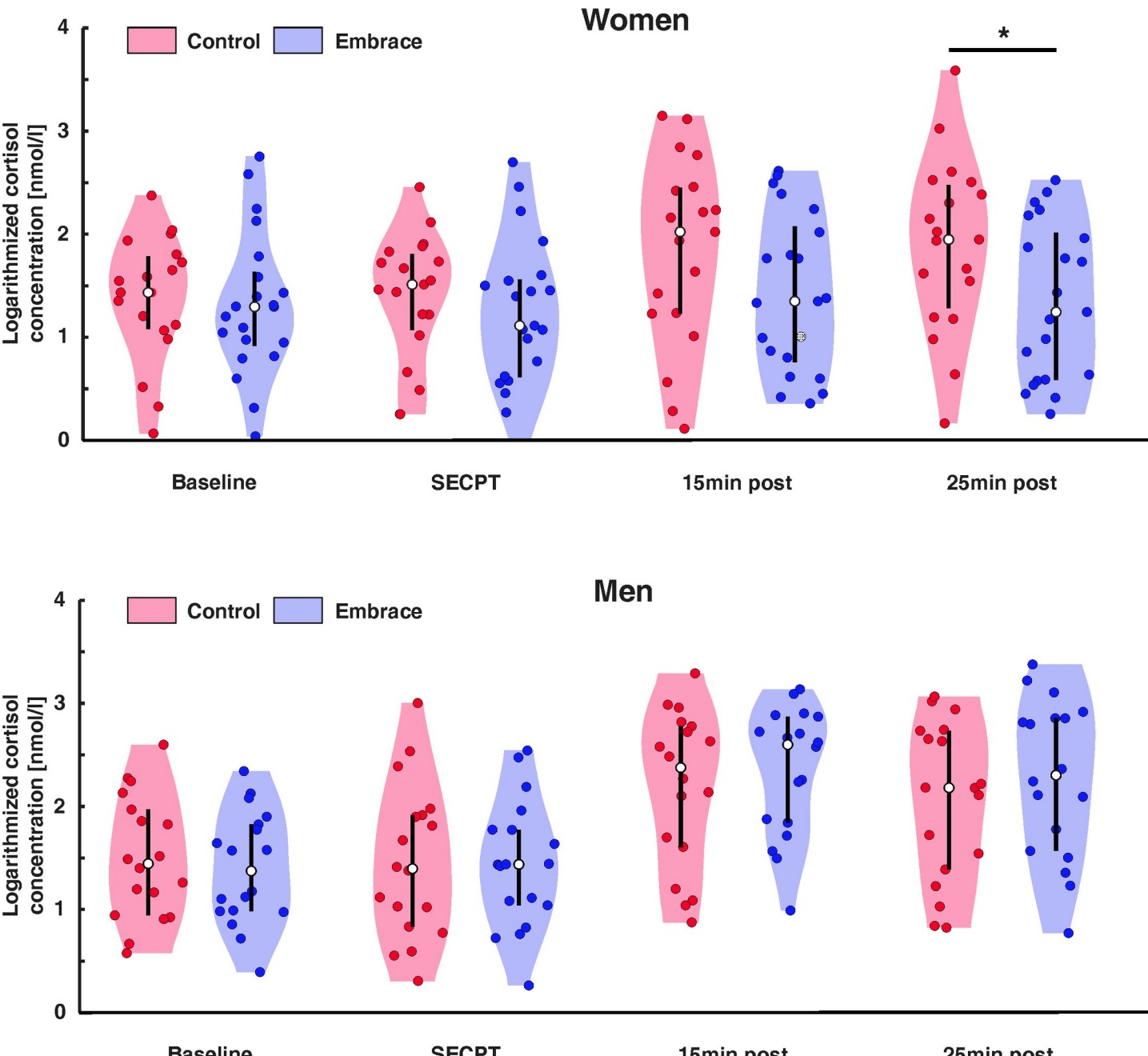

**Fig 2.** Cortisol concentration for women (top) and men (bottom) during baseline, the SECPT and 15 minutes as well as 25 minutes post SECPT for the embrace and control condition. White dots represent the median value for each group. Error bars represent the upper and lower quartiles. Group differences between the conditions are marked. $^*$ = $p < .05$.

corrected post hoc test revealed a significant increase in diastolic blood pressure during the SECPT compared to all other measurement time points (all $p$'s $< .001$, see S2 Fig). No other main effect or interaction reached significance (all $p$'s $> .068$). Descriptive values for diastolic blood pressure measurements across groups can be found in Table 3. The effect size of the model with and without interactions can be found in S1 Table.

**Table 1. Descriptive values (mean value and standard deviation) of logarithmized cortisol values across the four measurement time points broken down by condition and sex.**

| Time point | | Women | | | Men | |
|---|---|---|---|---|---|---|
| | Group | Mean | SD | Group | Mean | SD |
| Baseline | Control | 1.38 | .60 | Control | 1.50 | .60 |
| | Embrace | 1.32 | .69 | Embrace | 1.40 | .55 |
| | Group | Mean | SD | Group | Mean | SD |
| SECPT | Control | 1.39 | .62 | Control | 1.46 | .75 |
| | Embrace | 1.16 | .88 | Embrace | 1.49 | .62 |
| | Group | Mean | SD | Group | Mean | SD |
| 15min post | Control | 1.83 | .91 | Control | 2.18 | .76 |
| | Embrace | 1.42 | .76 | Embrace | 2.34 | .62 |
| | Group | Mean | SD | Group | Mean | SD |
| 25min post | Control | 1.89 | .84 | Control | 2.06 | .77 |
| | Embrace | 1.34 | .76 | Embrace | 2.28 | .77 |

## Affective state

For positive affective state, there was a significant main effect of measurement time point at the 15min and 25min post SECPT measurement ($\beta$ = -0.14 [-0.27 –-0.01], SE = 0.07, $t$ = 2.11, $p$ = .036; $\beta$ = -0.29 [-0.42 –-0.16], SE = 0.07, $t$ = 4.27, $p$ < .001, respectively). Bonferroni-corrected post-hoc tests however only reached significance for the 25min post SECPT measurement (see S3 Fig). Here, positive affect decreased with respect to baseline and the SECPT ($t$ = 4.27, $p$ < .001; $t$ = 3.88, $p$ < .001, respectively). No other main effect or interaction reached significance (all $p$'s > .227). Descriptive values for positive affective ratings across groups can be found in Table 4. The effect size of the model with and without interactions can be found in S1 Table.

For negative affective state, there was a significant main effect of measurement time point at the 25min post SECPT measurement ($\beta$ = -0.13 [-0.22 –-0.04], SE = 0.05, $t$ = 2.75, $p$ = .007, see S4 Fig). Bonferroni-corrected post-hoc tests revealed a significant decrease in negative affect at the 25min post SECPT measurement compared to the baseline and the SECPT itself ($t$ = 2.75, $p$ = .039; $t$ = 4.33, $p$ < .001). No other main effect or interaction reached significance

**Table 2. Descriptive values (mean value and standard deviation) of systolic blood pressure values across the four measurement time points broken down by condition and sex.**

| Time point | | Women | | | Men | |
|---|---|---|---|---|---|---|
| | Group | Mean | SD | Group | Mean | SD |
| Baseline | Control | 113.26 | 14.82 | Control | 129.72 | 14.71 |
| | Embrace | 114.95 | 14.60 | Embrace | 132.33 | 14.99 |
| | Group | Mean | SD | Group | Mean | SD |
| SECPT | Control | 125.37 | 12.98 | Control | 140.33 | 13.36 |
| | Embrace | 129.10 | 17.03 | Embrace | 146.61 | 17.22 |
| | Group | Mean | SD | Group | Mean | SD |
| 15min post | Control | 109.58 | 10.17 | Control | 121.06 | 10.81 |
| | Embrace | 110.33 | 14.14 | Embrace | 127.17 | 15.37 |
| | Group | Mean | SD | Group | Mean | SD |
| 25min post | Control | 106.53 | 10.10 | Control | 118.83 | 13.29 |
| | Embrace | 112.05 | 10.36 | Embrace | 124.06 | 14.73 |

**Table 3. Descriptive values (mean value and standard deviation) of diastolic blood pressure values across the four measurement time points broken down by condition and sex.**

| Time point | | Women | | | Men | |
|---|---|---|---|---|---|---|
| | Group | Mean | SD | Group | Mean | SD |
| Baseline | Control | 80.26 | 12.33 | Control | 77.94 | 12.39 |
| | Embrace | 82.67 | 12.93 | Embrace | 80.56 | 12.43 |
| | Group | Mean | SD | Group | Mean | SD |
| SECPT | Control | 91.16 | 8.40 | Control | 91.00 | 13.72 |
| | Embrace | 93.14 | 9.67 | Embrace | 96.78 | 13.26 |
| | Group | Mean | SD | Group | Mean | SD |
| 15min post | Control | 77.90 | 7.56 | Control | 75.11 | 13.54 |
| | Embrace | 81.24 | 14.86 | Embrace | 79.94 | 10.06 |
| | Group | Mean | SD | Group | Mean | SD |
| 25min post | Control | 78.47 | 8.65 | Control | 75.22 | 11.05 |
| | Embrace | 79.81 | 11.03 | Embrace | 78.50 | 8.21 |

(all $p$'s > .252). Descriptive values for negative affective ratings across groups can be found in Table 5. The effect size of the model with and without interactions can be found in S1 Table.

## Subjective SECPT ratings and SECPT duration

We neither found any main effects of condition or sex, nor any interaction for the difficulty, painfulness, stressfulness or unpleasantness ratings of the SECPT (all $p$'s > .059). The same was true for the time that the participants held their hands in the ice-cold water (all $p$'s > .051). Descriptive values for these measures across groups can be found in Table 6.

## Discussion

In the present study, we investigated the efficacy of a short-term embrace on the physiological stress response. To this end, we invited romantic couples and subjected them to the SECPT in a joint testing environment. One group was additionally instructed to embrace each other prior to the stress procedure. We hypothesized that both responses from the sympathetic nervous system as well as from the HPA axis would be decreased through the application of social

**Table 4. Descriptive values (mean value and standard deviation) of positive PANAS values across the four measurement time points broken down by condition and sex.**

| Time point | | Women | | | Men | |
|---|---|---|---|---|---|---|
| | Group | Mean | SD | Group | Mean | SD |
| Baseline | Control | 3.21 | .56 | Control | 3.06 | .55 |
| | Embrace | 3.03 | .57 | Embrace | 2.94 | .56 |
| | Group | Mean | SD | Group | Mean | SD |
| SECPT | Control | 3.23 | .75 | Control | 3.02 | .81 |
| | Embrace | 2.90 | .80 | Embrace | 3.00 | .69 |
| | Group | Mean | SD | Group | Mean | SD |
| 15min post | Control | 3.07 | .74 | Control | 2.84 | .65 |
| | Embrace | 2.99 | .89 | Embrace | 2.76 | .73 |
| | Group | Mean | SD | Group | Mean | SD |
| 25min post | Control | 2.90 | .58 | Control | 2.61 | .81 |
| | Embrace | 2.94 | .92 | Embrace | 2.61 | .76 |

**Table 5. Descriptive values (mean value and standard deviation) of negative PANAS values across the four measurement time points broken down by condition and sex.**

| Time point | | Women | | | Men | |
|---|---|---|---|---|---|---|
| | Group | Mean | SD | Group | Mean | SD |
| Baseline | Control | 1.26 | .19 | Control | 1.22 | .26 |
| | Embrace | 1.41 | .38 | Embrace | 1.32 | .42 |
| | Group | Mean | SD | Group | Mean | SD |
| SECPT | Control | 1.41 | .43 | Control | 1.27 | .40 |
| | Embrace | 1.45 | .39 | Embrace | 1.40 | .41 |
| | Group | Mean | SD | Group | Mean | SD |
| 15min post | Control | 1.20 | .31 | Control | 1.27 | .55 |
| | Embrace | 1.23 | .37 | Embrace | 1.24 | .29 |
| | Group | Mean | SD | Group | Mean | SD |
| 25min post | Control | 1.08 | .18 | Control | 1.21 | .58 |
| | Embrace | 1.24 | .32 | Embrace | 1.18 | .48 |

touch. We could partly confirm our hypothesis as there was a significantly reduced cortisol response in the embracing group compared to the control group. This effect was selective to women and could not be observed in men. Blood pressure as a marker of sympathetic nervous system activity and subjective ratings of affective state were not modulated by the embrace.

A noteworthy finding of our study is the sex difference in the influence of the embrace on cortisol changes in response to the SECPT. We found no evidence that men benefitted from a short-term embrace as a potential stress buffer and our results indicated that this effect is specific to women. The effect was not mediated by differences in relationship quality as there were no difference in relationship satisfaction between women and men. A conceivable explanation for this sex difference could relate to varying levels of oxytocin release between men and women following the embrace. Oxytocin is hypothesized to inhibit the synthesis of adrenocorticotropic hormone in the pituitary gland as it resembles vasopressin in its molecular structure [39]. Rising oxytocin levels are associated with decreases in vasopressin ultimately resulting in decreased cortisol secretion in the human body [39]. While we did not measure oxytocin in the present study, a meta-analytic study on sex differences in affective touch has demonstrated that women perceive affective touch as significantly more pleasant compared to men [40]. Moreover, oxytocin has been demonstrated to be directly correlated with the perceived pleasantness of gentle touch [41]. Therefore, the mutual embrace might have elicited higher levels of perceived pleasantness and thus higher levels of oxytocin release in women compared to men which could explain the observed difference. Another explanation could relate to the "tend-and-befriend" hypothesis which states that women respond differentially to stress-inducing events in the environment compared to men [42]. According to this hypothesis, women respond to stressful situations via an increase in care and protection for offspring, which in turn reduces the bodily stress response through an increase in oxytocin. This

**Table 6. Descriptive values (mean value and standard deviation) of the SECPT ratings and the duration of the SECPT broken down by condition and sex.**

| | Difficulty | Painfulness | Stressfulness | Unpleasantness | Duration in s |
|---|---|---|---|---|---|
| Embrace women | 69.05 ± 27.73 | 55.24 ± 32.03 | 44.29 ± 24.61 | 73.81 ± 25.39 | 148.0 ± 57.56 |
| Embrace men | 62.78 ± 24.49 | 45.56 ± 29.35 | 41.67 ± 23.33 | 62.78 ± 24.92 | 166.50 ± 39.32 |
| Control women | 70.35 ± 33.25 | 61.58 ± 33.54 | 54.74 ± 31.33 | 74.74 ± 27.56 | 143.68 ± 58.13 |
| Control men | 51.11 ± 30.66 | 54.44 ± 31.29 | 44.44 ± 24.79 | 67.78 ± 17.00 | 169.72 ± 34.91 |

evolutionary conserved mechanism to increase offspring survival could thus generalize to a variety of stressful situations. It should be noted however that a study investigating the oxytocin release in response to stress did not find marked sex differences [43]. Thus, this sex difference requires additional research.

Our results have implications for everyday life situations since increases in glucocorticoids have been shown to impair memory retrieval and executive functioning [44–47]. Memory traces show impaired retrieval under higher cortisol levels, especially if they have a negative valence [48–50]. Since everyday situations like an upcoming examination are associated with a pronounced cortisol increase [51], it is important to consider potential buffers against the negative effects of cortisol secretion on memory retrieval. In contrast to long massages or prolonged hand-holding as performed in previous studies [27, 28], a short-term embrace could be considered a highly feasible method in everyday life to buffer against these effects based on the findings of our study. It remains however an open question whether this effect is limited to romantic partner embraces or generalizes also to embraces between platonic friends. A recent survey study investigating the role of social touch on mental health aspects such as feelings of anxiety found that health benefits seem to be mostly linked to non-sexual intimate touch [52]. This finding would be in line with the observed effect in the study by Pauley et al. [29] in which relationship type moderated the cortisol reactivity in the participants. Future research is needed to further investigate this issue.

In contrast to our hypotheses, we did not observe any changes in sympathetic nervous system activity indicated by the absence of group difference in blood pressure, regardless of sex. This result is not in accordance with previous findings where prolonged exposure to social touch or affectionate communication resulted in reduced sympathetic nervous system activity indicated by decreased blood pressure and heart rate or increased heart rate variability [25, 27–29]. A possible explanation could relate to the pain associated with the SECPT as no other of the previously reported studies used a painful stimulus to induce stress. Pain strongly engages the sympathetic nervous system and is thus difficult to reduce [53]. Another possible explanation why we could not observe changes in blood pressure could relate to the time course of our experiment. The SECPT in our experiment took place immediately after the embrace. The very quick succession of the SECPT might have prevented the buffering effects of oxytocin to take effect as the blood concentration could have been too low at this point in time. Since this should have however also affected results for HPA-axis activation, this explanation is less likely.

Our study is subject to a few limitations that need to be acknowledged. First, while we controlled for OCs using them as covariates in the model, we did not collect data on the menstrual cycle in women that has been demonstrated to influence cortisol secretion [54]. Since our sample was randomly attributed to the experimental groups, it is however unlikely that there were systematic differences in cycle phases. Another limitation was the lack of an oxytocin measure which could have informed us about changes in oxytocin levels across the study, especially during the SECPT. It should be noted however that peripheral hematic and salivary oxytocin levels have been called into question whether they provide accurate measurements for oxytocin levels in the brain where the HPA axis is regulated [55]. A final limitation concerns the lack of an affective evaluation of the embrace since it could have illuminated whether men truly perceived the embrace as less pleasant compared to women.

In conclusion, we found a cortisol-buffering effect of embraces between romantic partners following a stress induction procedure. The effect was specific to women. This finding could have implications for stress reduction in everyday situations that often induce stress like exams, oral presentations or job interviews. Importantly, we also want to highlight our findings in the context of the COVID-19 pandemic that has substantially increased stress and

depression levels in many individuals due to economic and social restrictions [56, 57]. It could be conceived that these increases in everyday stress are in part due to the lack of social affective touch through the means of for example embraces. Future research needs to be conducted to identify the long-term impact of the pandemic on stress and social touch behavior.

## Supporting information

**S1 Table. Effect sizes of the models with and without interactions for cortisol, diastolic and systolic blood pressure as well as positive and negative affect.** Both $R^2$ and Cohen's f are given as effect size measures.
(DOCX)

**S1 Fig.** Systolic blood pressure for women (top) and men (bottom) during baseline, the SECPT and 15 minutes as well as 25 minutes post SECPT for the embrace and control condition. White dots represent the median value for each group. Error bars represent the upper and lower quartiles.
(TIF)

**S2 Fig.** Diastolic blood pressure for women (top) and men (bottom) during baseline, the SECPT and 15 minutes as well as 25 minutes post SECPT for the embrace and control condition. White dots represent the median value for each group. Error bars represent the upper and lower quartiles.
(TIF)

**S3 Fig.** Positive affect ratings for women (top) and men (bottom) during the baseline, SECPT and 15 minutes as well as 25 minutes post SECPT for the embrace and control condition. White dots represent the median value for each group. Error bars represent the upper and lower quartiles.
(TIF)

**S4 Fig.** Negative affect ratings for women (top) and men (bottom) during the baseline, SECPT and 15 minutes as well as 25 minutes post SECPT for the embrace and control condition. White dots represent the median value for each group. Error bars represent the upper and lower quartiles.
(TIF)

## Author Contributions

**Conceptualization:** Oliver T. Wolf, Sebastian Ocklenburg, Julian Packheiser.

**Data curation:** Gesa Berretz, Chantal Cebula, Blanca Maria Wortelmann, Panagiota Papadopoulou, Julian Packheiser.

**Formal analysis:** Gesa Berretz, Julian Packheiser.

**Funding acquisition:** Oliver T. Wolf, Sebastian Ocklenburg.

**Investigation:** Oliver T. Wolf, Julian Packheiser.

**Methodology:** Gesa Berretz, Chantal Cebula, Blanca Maria Wortelmann, Panagiota Papadopoulou, Sebastian Ocklenburg, Julian Packheiser.

**Project administration:** Sebastian Ocklenburg, Julian Packheiser.

**Resources:** Oliver T. Wolf.

**Software:** Julian Packheiser.

**Supervision:** Oliver T. Wolf, Sebastian Ocklenburg, Julian Packheiser.

**Validation:** Gesa Berretz.

**Visualization:** Gesa Berretz, Julian Packheiser.

**Writing – original draft:** Gesa Berretz, Julian Packheiser.

**Writing – review & editing:** Gesa Berretz, Chantal Cebula, Blanca Maria Wortelmann, Panagiota Papadopoulou, Oliver T. Wolf, Sebastian Ocklenburg, Julian Packheiser.

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
