## [Decision Letter · Decision Letter 0]

9 Feb 2022

PONE-D-21-38194Romantic partner embraces reduce cortisol release after acute stress induction in women but not in menPLOS ONE

Dear Dr. Packheiser,

Thank you for submitting your manuscript to PLOS ONE. After careful consideration, we feel that it has merit but does not fully meet PLOS ONE’s publication criteria as it currently stands. Therefore, we invite you to submit a revised version of the manuscript that addresses the points raised during the review process.

Please comment about power analysis for sample size

Discuss more in detail gender difference

Please quote pandemic restrictions effects

Please clarify the difference between romantic partner embrace and other types such as parents, friends, etc.

We look forward to receiving your revised manuscript.

Kind regards,

Marta Panzeri, Ph.D.

Academic Editor

PLOS ONE

Journal Requirements:

3. We noted in your submission details that a portion of your manuscript may have been presented or published elsewhere. 

( The results have been pre-published at https://psyarxiv.com/32bde/. It is not under consideration elsewhere.)

Reviewers' comments:

Reviewer's Responses to Questions

**Comments to the Author**

1. Is the manuscript technically sound, and do the data support the conclusions?

Reviewer #1: Yes

Reviewer #2: Yes

2. Has the statistical analysis been performed appropriately and rigorously? 

Reviewer #1: Yes

Reviewer #2: Yes

3. Have the authors made all data underlying the findings in their manuscript fully available?

Reviewer #1: Yes

Reviewer #2: No

4. Is the manuscript presented in an intelligible fashion and written in standard English?

Reviewer #1: Yes

Reviewer #2: Yes

5. Review Comments to the Author

Reviewer #1: This is a very interesting study adjudicating the effects of a brief affectionate embrace on physiological reactivity to cold pressor. In my opinion, the study is well designed and executed, and I recommend publication after attention to a few small issues.

Given that the sample comprised heterosexual couples, it is confusing that the numbers of women and men are not equal, until the authors explain that the imbalance “resulted from posthoc exclusions of participants due to failure to meet the inclusion criteria or empty salivettes.” I understand the issue of empty salivettes, but how were participants enrolled in the study in the first place if they didn’t meet inclusion criteria?

I think the authors are overlooking a few relevant studies. For instance, Grewen and colleagues used a similar experimental design but examined the effects of handholding rather than hugging, and Pauley et al. used a similar manipulation with a variety of stress inducers.

This is a fairly small N, so I wonder how the authors arrived at this sample size? Was an a-priori power analysis conducted? Even a post-hoc sensitivity analysis would be useful to justify the sample size.

Regarding the authors’ post-hoc explanation of their unexpected sex difference—and regarding their implication of cortisol, specifically—they might find value in Shelley Taylor’s tend-and-befriend hypothesis, which similarly explains why affectionate behavior may benefit women more than it benefits men (although they should also note Floyd et al.’s experiment, which did not find that sex difference in oxytocinergic reactions).

Floyd, K., Pauley, P. M., & Hesse, C. (2010). State and trait affectionate communication buffer adults’ stress reactions. Communication Monographs, 77(4), 618–636. https://doi.org/10.1080/03637751.2010.498792

Grewen, K. M., Anderson, B. J., Girdler, S. S., & Light, K. C. (2003). Warm partner contact is related to lower cardiovascular reactivity. Behavioral Medicine, 29(3), 123–130. https://doi.org/10.1080/08964280309596065

Pauley, P. M., Floyd, K., & Hesse, C. (2015). The stress-buffering effects of a brief dyadic interaction before an acute stressor. Health Communication, 30(7), 646–659. https://doi.org/10.1080/10410236.2014.888385

Taylor, S. E., Klein, L. C., Lewis, B. P., Gruenewald, T. L., Guring, R. A. R., & Updegraff, J. A. (2000). Biobehavioral responses to stress in females: Tend-and-befriend, not fight-or-flight. Psychological Review, 107(3), 411–429. https://doi.org/10.1037/0033-295X.107.3.411

Reviewer #2: Review of the manuscript entitled “Romantic partner embraces reduce cortisol release after acute stress induction in women but not in men” by Berretz and colleagues.

In this interesting study, the authors tested the effect of a short-term embrace between romantic partners on the subjective/objective measurements of acute stress. They divided 76 participants in a control and in an experimental group, and they induced acute stress via the Socially Evaluated Cold Pressor Test (SECPT). Stress was measured by means of: i) cortisol response, ii) sympathetic response (blood pressures), and iii) subjective affect ratings (PANAS questionnaire). All these measures were collected 1) at baseline, 2) SECPT, 3) 15 minutes after SECPT, 4) 25 minutes after SECPT. Results revealed a reduced cortisol response in the embracing group compared to the control group, but only in women. Neither blood pressure nor subjective ratings were modulated by the embrace.

The study is well-conceived and the manuscript is clear and well-written. Results are original and of interest for both specialists and for a wider audience. I have not major concerns about this manuscript, but only some minor hints that could help the authors to further improve the clarity of the text.

1) Page 5: “The other group did not embrace each other, and the partners only provided social support during the joint testing procedure”. Can the authors better explain what is meant here with “social support”? At a first glance, this is not clear.

2) In the Participants section it is stated that 36 participants were male and 40 were female, but that all participants were in heterosexual relationships. In the following paragraph this point is clarified, but at this point it sounds strange: I suggest explaining here that some participants were excluded because of technical issues.

3) In the same section, the sample’s BMI is given, but I do not understand the reason why it can be useful. Can the authors justify this information? Alternatively, maybe it can be deleted.

4) Did the authors ask to participants the duration of their relationship? Can this info alter the effect of the embrace? In other words, can we expect a different effect of the embrace in accordance with the duration of the relationship (e.g., stronger effect in long-lasting relations?).

5) Page 8: “Following the SECPT, the couples were separated from each other to fill out the Relationship Assessment Scale (RAS) in an unbiased manner”. Please, explain the “unbiased manner”: what does it mean?

6) From a theoretical point of view, can we expect different results according to the specific person involved in the embrace? Specifically, is the embrace of the romantic partner similar to that of the mother? Can we expect different cortisol effect according to the “role” of the persons? What can we expect of the embrace is carried out by an unknown person? In other words, is the embrace per se or the subjective experience of embracing the partner responsible for the effect found here? This point should be discussed, at least in the final section of the manuscript.

7) In the last sentence, I would also add a reference to the pandemic: reading the manuscript, I think to different everyday situations, such as exams, oral presentations or job interviews – as listed by the authors – but also to the importance of the physical contact in this period of social distancing. I would like to suggest inserting this crucial applied implication of this result.

6. PLOS authors have the option to publish the peer review history of their article (what does this mean?). If published, this will include your full peer review and any attached files.

Reviewer #1: No

Reviewer #2: **Yes: **Giulia Prete

---

## [Author Response · Author response to Decision Letter 0]

9 Mar 2022

Response to Reviews

Comments by the editor:

Dear Dr. Packheiser,

Thank you for submitting your manuscript to PLOS ONE. After careful consideration, we feel that it has merit but does not fully meet PLOS ONE’s publication criteria as it currently stands. Therefore, we invite you to submit a revised version of the manuscript that addresses the points raised during the review process. 

Please comment about power analysis for sample size

Discuss more in detail gender difference

Please quote pandemic restrictions effects

Please clarify the difference between romantic partner embrace and other types such as parents, friends, etc.

We look forward to receiving your revised manuscript.

Kind regards,

Marta Panzeri, Ph.D.

Academic Editor

PLOS ONE

Response:

We thank the editor for her positive assessment of our manuscript. Attached you will find our response letter in which we address the concerns of the reviewer point-by-point. The changes made in accordance with the reviewers are highlighted in the marked manuscript. If longer passages have been added to the manuscript, we explicitly state them in the response letter. We explicitly also want to thank both reviewers for their helpful comments that significantly improved the manuscript conceptually and heightened its relevance with respect to the ongoing pandemic. An important note to the reviewers is that we changed the citation style in the manuscript in accordance with the PloS One guidelines. However, we chose to use non-numbered citations when we referenced the relevant sections in the response letter to facilitate what additional research has been incorporated in the manuscript.

Reviewer comments:

Reviewer #1: 

This is a very interesting study adjudicating the effects of a brief affectionate embrace on physiological reactivity to cold pressor. In my opinion, the study is well designed and executed, and I recommend publication after attention to a few small issues.

Response:

We thank the reviewer for their positive evaluation and helpful comments on our study.

Point 1:

Given that the sample comprised heterosexual couples, it is confusing that the numbers of women and men are not equal, until the authors explain that the imbalance “resulted from posthoc exclusions of participants due to failure to meet the inclusion criteria or empty salivettes.” I understand the issue of empty salivettes, but how were participants enrolled in the study in the first place if they didn’t meet inclusion criteria?

Response:

We agree with the reviewer that this must have seemed confusing as both reviewers stumbled over this particular paragraph. In accordance with the second reviewer, we re-structured this paragraph so that the information about the exclusion due to technical issues appears earlier in the manuscript. With regard to the post-hoc exclusion of participants, contact was in most cases only established with one romantic partner during recruitment. Thus, on rare occasions, the other partner was not informed about all relevant inclusion criteria before arriving at the study location. In these cases, we decided to nonetheless move forward with the study protocol as at least one participant was meeting the inclusion criteria. The other partner was excluded. In accordance with the second reviewer, we simply state that the numerical imbalance was due to technical issues as this detailed explanation seems unnecessary for the reader.

It reads:

“The numerical imbalance in the groups resulted from post hoc exclusion of participants due to technical issues.”

Point 2:

I think the authors are overlooking a few relevant studies. For instance, Grewen and colleagues used a similar experimental design but examined the effects of handholding rather than hugging, and Pauley et al. used a similar manipulation with a variety of stress inducers.

Response:

We are grateful that the reviewer has pointed us towards these extremely relevant studies. We added information on both studies to the introduction and discussion. The section in the introduction reads:

“In line with these findings, Grewen, Anderson, Girdler, and Light (2003) compared cohabitating couples that either held hands for a prolonged period (10min) and shortly embraced each other afterwards (20s) or did not touch each other prior to a public-speaking task. They found that the mutual physical contact attenuated systolic, diastolic blood pressure and heart rate increases compared to the control group. Finally, Pauley, Floyd, and Hesse (2015) subjected romantic couples or platonic friends to acute stress after they talked either about fond mutual memories for a period of 10 minutes and embraced afterwards (< 10s duration), only shared each other’s presence or waited separately for the stress induction to begin. The authors observed an attenuated cardiovascular stress response in all participants that went through the affectionate communication plus embracing condition. For HPA-axis activation, the findings were moderated by the relationship type as platonic friends showed a stronger increase in cortisol after expressing mutual affection compared to romantic partners that only shared each other’s presence.”

The sections in the discussion read:

“It remains however an open question whether this effect is limited to romantic partner embraces or generalizes also to embraces between platonic friends. A recent survey study investigating the role of social touch on mental health aspects such as feelings of anxiety found that health benefits seem to be mostly linked to non-sexual intimate touch (Mohr, Kirsch, & Fotopoulou, 2021). This finding would be in line with the observed effect in the study by Pauley et al. (2015) in which relationship type moderated the cortisol reactivity in the participants. Future research is needed to further investigate this issue.

In contrast to our hypotheses, we did not observe any changes in sympathetic nervous system activity indicated by the absence of group difference in blood pressure, regardless of sex. This result is not in accordance with previous findings where prolonged exposure to social touch or affectionate communication resulted in reduced sympathetic nervous system activity indicated by decreased blood pressure and heart rate or increased heart rate variability (Ditzen et al., 2007; Grewen et al., 2003; Meier et al., 2020; Pauley et al., 2015). A possible explanation could relate to the pain associated with the SECPT as no other of the previously reported studies used a painful stimulus to induce stress. Pain strongly engages the sympathetic nervous system and is thus difficult to reduce (Schlereth & Birklein, 2008).”

Mohr, M. von, Kirsch, L. P., & Fotopoulou, A. (2021). Social touch deprivation during COVID-19: Effects on psychological wellbeing and craving interpersonal touch. Royal Society Open Science, 8(9), 210287. https://doi.org/10.1098/rsos.210287

Schlereth, T., & Birklein, F. (2008). The sympathetic nervous system and pain. NeuroMolecular Medicine, 10(3), 141–147. https://doi.org/10.1007/s12017-007-8018-6

Point 3:

This is a fairly small N, so I wonder how the authors arrived at this sample size? Was an a-priori power analysis conducted? Even a post-hoc sensitivity analysis would be useful to justify the sample size.

Response:

We agree with the reviewer that the sample could have been extended. The limited sample size was largely due to difficulties in data acquisition during the pandemic. While we did not conduct an a priori power analysis, we conducted a post hoc sensitivity analysis in g*Power in accordance with the reviewer’s suggestion. First, we determined the effect size of the three-way interaction in the model by subtracting the R² value of a model containing only main effects from the full interaction model. The effect size of the interaction alone was at R² = 0.042 or Cohen’s f = 0.21. A sensitivity analysis was conducted in g*Power (Faul et al., 2007) in a repeated measures within-between subject interaction model containing four groups (embracing, control, male, female) and four measurements. Correlations between measures were on average high (r = 0.7) and sphericity was violated (Greenhouse-Geisser ε = 0.62). Using these settings, 80% power would have been achieved with an effect of Cohen’s f = 0.14. We added a paragraph in the methods section that reads:

“A post hoc sensitvity analysis was conducted to determine how much power would have been necessary given the present sample size to detect within-between interaction effect using g*Power (Faul, Erdfelder, Lang, & Buchner, 2007). Here, we used four groups, four repeated measures, a repeated measures correlation of r = 0.7 and non-sphericity correction of ε = 0.62 as inputs. 80% power would have been achieved at interaction effects greater than Cohen’s f = 0.14.”

 We added the effect size for the three-way interactions of each dependent variable that was evaluated via a mixed-model to the supplementary information.

Supplementary table 1. Effect sizes of the models with and without interactions for cortisol, diastolic and systolic blood pressure as well as positive and negative affect. Both R² and Cohen’s f are given as effect size measures.

 Cortisol Systolic BP Diastolic BP Positive Affect Negative Affect

Effect size without interactions R² = 0.164

f = 0.44 R² = 0.388

f = 0.80 R² = 0.237

f = 0.56 R² = 0.038

f = 0.20 R² = 0.049

f = 0.23

Effect size with interactions

 R² = 0.206

f = 0.51 R² = 0.388

f = 0.80 R² = 0.239

f = 0.56 R² = 0.055

f = 0.24 R² = 0.062

f = 0.26

Interaction effect alone R² = 0.042

f = 0.21 R² = 0

f = 0 R² = 0.002

f = 0.04 R² = 0.017

f = 0.13 R² = 0.013

f = 0.11

Faul, F., Erdfelder, E., Lang, A.‑G., & Buchner, A. (2007). G* Power 3: A flexible statistical power analysis program for the social, behavioral, and biomedical sciences. Behavior Research Methods, 39(2), 175–191.

We also estimated the effect size of the relevant comparison between the female embracing and control group in the linear mixed model using the eff_size function from the emmeans package. This way, random effects and covariates are accounted for in the effect size estimation. Here, a large effect size was estimated (Cohen’s f = 0.62, standard error = 0.27). Using this effect size in a simple independent sample t-test model revealed 96.6% power to detect the effect. We are therefore confident that the employed sample size was sufficient for the present study. We hope that the reviewer agrees with our assessment. 

Point 4:

Regarding the authors’ post-hoc explanation of their unexpected sex difference—and regarding their implication of cortisol, specifically—they might find value in Shelley Taylor’s tend-and-befriend hypothesis, which similarly explains why affectionate behavior may benefit women more than it benefits men (although they should also note Floyd et al.’s experiment, which did not find that sex difference in oxytocinergic reactions).

Response:

We thank the reviewer for pointing this out as this hypothesis provides a further possible explanation for the observed sex difference. We added the suggested literature to the discussion. The section reads:

“Another explanation could relate to the “tend-and-befriend” hypothesis which states that women respond differentially to stress-inducing events in the environment compared to men (Taylor et al., 2000). According to this hypothesis, women respond to stressful situations via an increase in care and protection for offspring, which in turn reduces the bodily stress response through an increase in oxytocin. This evolutionary conserved mechanism to increase offspring survival could thus generalize to a variety of stressful situations. It should be noted however that a study investigating the oxytocin release in response to stress did not find marked sex differences (Floyd et al., 2010). Thus, this sex difference requires additional research.” 

Floyd, K., Pauley, P. M., & Hesse, C. (2010). State and trait affectionate communication buffer adults’ stress reactions. Communication Monographs, 77(4), 618–636. https://doi.org/10.1080/03637751.2010.498792

Grewen, K. M., Anderson, B. J., Girdler, S. S., & Light, K. C. (2003). Warm partner contact is related to lower cardiovascular reactivity. Behavioral Medicine, 29(3), 123–130. https://doi.org/10.1080/08964280309596065

Pauley, P. M., Floyd, K., & Hesse, C. (2015). The stress-buffering effects of a brief dyadic interaction before an acute stressor. Health Communication, 30(7), 646–659. https://doi.org/10.1080/10410236.2014.888385

Taylor, S. E., Klein, L. C., Lewis, B. P., Gruenewald, T. L., Guring, R. A. R., & Updegraff, J. A. (2000). Biobehavioral responses to stress in females: Tend-and-befriend, not fight-or-flight. Psychological Review, 107(3), 411–429. https://doi.org/10.1037/0033-295X.107.3.411

Reviewer #2: 

Review of the manuscript entitled “Romantic partner embraces reduce cortisol release after acute stress induction in women but not in men” by Berretz and colleagues.

In this interesting study, the authors tested the effect of a short-term embrace between romantic partners on the subjective/objective measurements of acute stress. They divided 76 participants in a control and in an experimental group, and they induced acute stress via the Socially Evaluated Cold Pressor Test (SECPT). Stress was measured by means of: i) cortisol response, ii) sympathetic response (blood pressures), and iii) subjective affect ratings (PANAS questionnaire). All these measures were collected 1) at baseline, 2) SECPT, 3) 15 minutes after SECPT, 4) 25 minutes after SECPT. Results revealed a reduced cortisol response in the embracing group compared to the control group, but only in women. Neither blood pressure nor subjective ratings were modulated by the embrace. 

The study is well-conceived and the manuscript is clear and well-written. Results are original and of interest for both specialists and for a wider audience. I have not major concerns about this manuscript, but only some minor hints that could help the authors to further improve the clarity of the text.

Response:

We thank the reviewer for her positive and constructive feedback on our study.

Point 1:

Page 5: “The other group did not embrace each other, and the partners only provided social support during the joint testing procedure”. Can the authors better explain what is meant here with “social support”? At a first glance, this is not clear.

Response:

We thank the reviewer for pointing this out. The couples were not prohibited to talk to each other prior to the SECPT in the either group. We changed this sentence to provide clarification on this matter, as we did not specify how the support was constituted. It reads: “The other group did not embrace each other and partners only provided support through their physical presence.” 

Point 2:

In the Participants section it is stated that 36 participants were male and 40 were female, but that all participants were in heterosexual relationships. In the following paragraph this point is clarified, but at this point it sounds strange: I suggest explaining here that some participants were excluded because of technical issues.

Response:

Both reviewers had an issue with this description. We explained the underlying technical issues under point 1 for reviewer 1. In accordance with your comments, we now state that participants were excluded due to technical issues in the participants section.

Point 3:

In the same section, the sample’s BMI is given, but I do not understand the reason why it can be useful. Can the authors justify this information? Alternatively, maybe it can be deleted.

Response:

Obesity has been linked to a systematic increase in HPA-axis responsivity. Thus, participants with considerable overweight were excluded from the study. We added this information to the manuscript. It reads:

“Furthermore, the body mass index was between 18.5 and 27 since obesity has been linked to systematic increases in HPA-axis responsivity (Rodriguez et al., 2015).”

Rodriguez, A. C. I., Epel, E. S., White, M. L., Standen, E. C., Seckl, J. R., & Tomiyama, A. J. (2015). Hypothalamic-pituitary-adrenal axis dysregulation and cortisol activity in obesity: a systematic review. Psychoneuroendocrinology, 62, 301-318.

Point 4:

Did the authors ask to participants the duration of their relationship? Can this info alter the effect of the embrace? In other words, can we expect a different effect of the embrace in accordance with the duration of the relationship (e.g., stronger effect in long-lasting relations?).

Response:

Information on relationship duration was unfortunately not assessed. While it could be a possible moderator in the present study, we believe that it is more likely that the relationship satisfaction would play a more direct role to the embracing effect. We explored this possibility and added the RAS score to the model as a further variable. The results were unchanged, but relationship satisfaction did not reach significance indicating that it did not play a major role in the effect of the embrace.

Point 5:

Page 8: “Following the SECPT, the couples were separated from each other to fill out the Relationship Assessment Scale (RAS) in an unbiased manner”. Please, explain the “unbiased manner”: what does it mean? 

Response:

We rephrased this sentence to be clearer. We meant that the participants would have been likely influenced by the presence of their partner and would have likely avoided negative ratings of the relationship. We clarified this in the revised manuscript. It reads:

“This was done to prevent any influence on the rating by the romantic partners by having to fill out the questionnaire next to them.”

Point 6:

From a theoretical point of view, can we expect different results according to the specific person involved in the embrace? Specifically, is the embrace of the romantic partner similar to that of the mother? Can we expect different cortisol effect according to the “role” of the persons? What can we expect of the embrace is carried out by an unknown person? In other words, is the embrace per se or the subjective experience of embracing the partner responsible for the effect found here? This point should be discussed, at least in the final section of the manuscript.

Response:

The reviewer raises a very interesting point here. Indeed, the relationship between the embracing individuals is likely to play a role in these experiments as a recent survey study has demonstrated that only intimate (non-sexual) social touch can for example decreases feelings of anxiety and generally increase psychological well-being (Mohr et al., 2021). Friendly and professional touch did not provide a similar effect. Thus, the observed effect in our study seems to be limited to either romantic partners or close friends. We added a paragraph on this in the discussion. 

It reads:

“It remains however an open question whether this effect is limited to romantic partner embraces or generalizes also to embraces between platonic friends. A recent survey study investigating the role of social touch on mental health aspects such as feelings of anxiety found that health benefits seem to be mostly linked to non-sexual intimate touch (Mohr, Kirsch, & Fotopoulou, 2021). This finding would be in line with the observed effect in the study by Pauley et al. (2015) in which relationship type moderated the cortisol reactivity in the participants. Future research is needed to further investigate this issue.”

Point 7:

In the last sentence, I would also add a reference to the pandemic: reading the manuscript, I think to different everyday situations, such as exams, oral presentations or job interviews – as listed by the authors – but also to the importance of the physical contact in this period of social distancing. I would like to suggest inserting this crucial applied implication of this result.

Response:

We fully agree with the reviewer that the pandemic should be mentioned given the importance of social touch in physical and mental well-being and the current restrictions and social distancing mandate. We added this to the last paragraph.

It reads:

“Importantly, we also want to highlight our findings in the context of the COVID-19 pandemic that has substantially increased stress and depression levels in many individuals due to economic and social restrictions (Ettman et al., 2020; Salari et al., 2020). It could be conceived that these increase in everyday stress are in part to the lack of social affective touch through the means of for example embraces. Future research needs to be conducted to identify the long-term impact of the pandemic on stress and social touch behavior.”

---

## [Decision Letter · Decision Letter 1]

30 Mar 2022

Romantic partner embraces reduce cortisol release after acute stress induction in women but not in men

PONE-D-21-38194R1

Dear Dr. Packheiser,

We’re pleased to inform you that your manuscript has been judged scientifically suitable for publication and will be formally accepted for publication once it meets all outstanding technical requirements.

Kind regards,

Marta Panzeri, Ph.D.

Academic Editor

PLOS ONE

Additional Editor Comments (optional):

Reviewers' comments:

Reviewer's Responses to Questions

**Comments to the Author**

1. If the authors have adequately addressed your comments raised in a previous round of review and you feel that this manuscript is now acceptable for publication, you may indicate that here to bypass the “Comments to the Author” section, enter your conflict of interest statement in the “Confidential to Editor” section, and submit your "Accept" recommendation.

Reviewer #1: All comments have been addressed

Reviewer #2: (No Response)

2. Is the manuscript technically sound, and do the data support the conclusions?

Reviewer #1: (No Response)

Reviewer #2: Yes

3. Has the statistical analysis been performed appropriately and rigorously? 

Reviewer #1: (No Response)

Reviewer #2: Yes

4. Have the authors made all data underlying the findings in their manuscript fully available?

Reviewer #1: (No Response)

Reviewer #2: No

5. Is the manuscript presented in an intelligible fashion and written in standard English?

Reviewer #1: (No Response)

Reviewer #2: Yes

6. Review Comments to the Author

Reviewer #1: (No Response)

Reviewer #2: I would like to congratulate with the authors for their great work! They fully considered all the suggestions received and I believe the present version of the manuscript is now clearer and complete. I just want to suggest a couple minor hints which can be helpful in providing further support for the conclusions of the present study.

Firstly, concerning the explanation of the sex difference, a reference could also be made to different results showing that females are more “responsive” than males to “social stimuli” in general, which can be intended as in line with the results found here (see for instance https://pubmed.ncbi.nlm.nih.gov/28175962/
https://pubmed.ncbi.nlm.nih.gov/19083993/
https://pubmed.ncbi.nlm.nih.gov/18461176/).

Secondly, concerning the importance of the present results in the light of the pandemic, I want to indicate a study carried out *during* the quarantine imposed by the government at the beginning of the pandemic, revealing that persons who were highly fearful for the COVID-19 (higher worry), were objectively the most anxious as measured by means of standardized tests (https://pubmed.ncbi.nlm.nih.gov/33362631/). This is in line with the conclusions proposed by the authors and can be added to further support this view.

Finally, I want to express a special mention for the effort made to add all the new statistical evidence, which surely increases the completeness of the manuscript!

7. PLOS authors have the option to publish the peer review history of their article (what does this mean?). If published, this will include your full peer review and any attached files.

Reviewer #1: No

Reviewer #2: No

---

## [Editor Report · Acceptance letter]

26 Apr 2022

PONE-D-21-38194R1 

Romantic partner embraces reduce cortisol release after acute stress induction in women but not in men 

Dear Dr. Packheiser:

I'm pleased to inform you that your manuscript has been deemed suitable for publication in PLOS ONE. Congratulations! Your manuscript is now with our production department. 

Kind regards, 

on behalf of

Dr. Marta Panzeri 

Academic Editor

PLOS ONE